# Machine-Guided Design of Oxidation-Resistant Superconductors for Quantum Information Applications

**Carson Koppel [1,\*], Brandon Wilfong [2,3], Allana Iwanicki [2,3], Elizabeth Hedrick [3,4], Tanya Berry [5] and Tyrel M. McQueen [2,3,4,\*]**

1   Department of Physics and Astronomy, SUNY Stony Brook, Stony Brook, NY 11794, USA
2   Department of Chemistry, The Johns Hopkins University, Baltimore, MD 21218, USA
3   Institute for Quantum Matter, William H. Miller III Department of Physics and Astronomy, The Johns Hopkins University, Baltimore, MD 21218, USA
4   Department of Materials Science and Engineering, The Johns Hopkins University, Baltimore, MD 21218, USA
5   Department of Chemistry, Princeton University, Princeton, NJ 08540, USA
\*   Correspondence: carson.koppel@stonybrook.edu (C.K.); mcqueen@jhu.edu (T.M.M.)

**Abstract:** Decoherence in superconducting qubits has long been attributed to two-level systems arising from the surfaces and interfaces present in real devices. A recent significant step in reducing decoherence was the replacement of superconducting niobium by superconducting tantalum, resulting in a tripling of transmon qubit lifetimes ($T_1$). The identity, thickness, and quality of the native surface oxide, is thought to play a major role, as tantalum only has one oxide whereas niobium has several. Here we report the development of a thermodynamic metric to rank materials based on their potential to form a well-defined, thin, surface oxide. We first computed this metric for known binary and ternary metal alloys using data available from the Materials Project and experimentally validated the strengths and limits of this metric through the preparation and controlled oxidation of eight known metal alloys. Then we trained a convolutional neural network to predict the value of this metric from atomic composition and atomic properties. This allowed us to compute the metric for materials that are not present in the Materials Project, including a large selection of known superconductors, and, when combined with $T_c$, allowed us to identify new candidate superconductors for quantum information science and engineering (QISE) applications. We tested the oxidation resistance of a pair of these predictions experimentally. Our results are expected to lay the foundation for the tailored and rapid selection of improved superconductors for QISE.

**Keywords:** superconductivity; materials design; quantum science





## 1. Introduction

The field of quantum information science and engineering (QISE) has exploded over the past decade, with rapid advances in quantum sensors [1,2], quantum networks [3,4], quantum transduction [5,6], and quantum computing technologies [7,8]. Integral to many of these advances are incorporation of new and improved crystalline materials, whether as hosts for single-site quantum bits (qubits) [9–12], or as the active element in many-body quantum devices, e.g., the replacement of niobium with tantalum in superconducting transmon qubits [13].

For QISE to advance beyond the noisy intermediate-scale quantum (NISQ) era, step change improvements in qubit coherence times and gate fidelities are required. Much like the computing revolution, where vacuum tube computing logic was superseded by the transistor following the development of new, ultrapure, materials, materials design is needed to make step change advances within QISE. Prior studies seeking to guide materials selection within QISE have focused on utilizing a combination of computational tools (e.g., DFT or GW methods), chemical intuition, and Edisonian experimental mapping of parameter space to screen through the large number of known materials [9,10,12,13]. For

example, data mining efforts were used to identify all even-element-containing compounds that might serve as low nuclear spin background hosts for quantum defects [9], and a combination of high throughput and high-fidelity computations were used to identify host materials for ultralong $T_2$ coherence times on single-site defects [10,12]. Chemical intuition that the plethora of conducting niobium oxides that exist might be limiting transmon performance spurred a switch to tantalum, which does not suffer such issues, and resulted in a ~3× improvement in transmon lifetimes [13].

Aluminum, which superconducts below 1.2 K, was one of the first materials used to build superconducting qubits, due to its compatibility with microelectronic fabrication processes, and the intrinsic stability of a thin, robust, native oxide, making highly reproducible Josephson junctions [14]. However, in the case of transmon qubits, it was found that using a higher temperature superconductor—initially niobium and more recently tantalum—for the resonator portion was advantageous (the Josephson junction continues to be made of aluminum/aluminum oxide/aluminum) [13,15,16]. Numerous studies have focused on trying to understand the microscopic material mechanisms behind the 3× improvement in transmon lifetime when tantalum is substituted for niobium [17,18]. Detailed correlation of resonator performance with measurements of the surface oxide reveal that a uniform composition and thin conformal coating correlates with high-quality factors. Using dry-, rather than wet-etch processes results in a further 1.5–2× improvement in device lifetime, again attributed to a thinner oxide and more atomically precise interface with the superconducting metal [19].

The findings that the interfaces between the superconductor and metal oxide remain a significant source of decoherence motivates searches for new superconductors to further suppress this loss mechanism, including not just resonator but also the Josephson junction elements. This is especially important given recent work [18] that suggests the interfaces between superconductors in transmons, i.e., between the tantalum and the aluminum used to prepare the Josephson junctions (JJs) of full devices, is also a source of loss, so finding a superconductor that can replace both tantalum and aluminum could bring multiplicative benefits to performance.

Designing such improvements with computational-based tools is, however, challenging, because it is not currently possible to reliably predict the superconducting properties of materials [20,21] and because oxide formation on a materials surface is a complex process that depends very sensitively on nano- and micro-scale structural details [22,23]. Thus, materials identification has been limited to manual inspection of lists of known superconductors and cross-referencing with reported studies of oxide formation. This process identifies Mo–Re alloys [24,25] as potential candidates for next generation transmon devices. Unfortunately, the critical temperatures ($T_c$'s) are low, and the extreme melting points present practical processing challenges.

Here we report the development of a thermodynamic metric to rank a material's quality with respect to the formation of surface oxides. Computing this metric requires knowledge of the heats of formation of elements, alloys, and metal oxides, which can be obtained either experimentally [26] or computationally [27]. We assess the quality of this metric by experimentally preparing a set of metal alloys and carrying out assessment of oxidation under controlled conditions. We then trained a convolutional neural network (CNN) to predict the value of this metric solely from atomic properties and stoichiometries in order to be able to use this to screen superconductors because most of the thermodynamic information for these materials is unavailable. By combining with reported $T_c$'s, we then identified superconductors with a high figure of merit, with experimental validation of oxidation resistance trends.

## 2. Methods

The program to calculate the metric provided by Equation (6) from computational data in the Materials Project [28] was written in Python as a Jupyter Notebook. Python version 3.10.5 was used, and we used data from the Materials Project collected from 4 July 2022 to

18 July 2022. The calculations were produced as follows: first a list of metal elements was created, excluding transuranics and metalloids. Each metal was combined with another and then the resulting alloy searched for by name in the Materials Project (through its API), which returned that alloy and all its variants. Among the list of variants, the first that had the property of being experimentally observed rather than being computationally predicted only was chosen. This process was repeated until a list of 1562 binary alloys was generated. Then for each metal in the original list mentioned, the oxide with the highest stable oxidation state was selected. The program retrieved all the oxides of each element (by searching for all compounds in the database with a chemical formula, including the element in question and oxygen) and then calculated their oxidation states and selected the oxide with the highest one. Only chemically feasible and experimentally observed oxides were selected.

Then, for each alloy, the oxides of its constituent elements were identified and the metric was calculated. For example, for TiNb, its oxides are $TiO_2$ and $Nb_2O_5$. The heat of formation was calculated for the oxides if they were formed from the alloy: $TiNb + O_2$ or a naive mixture: $Ti + Nb + O_2$. The program balances the reaction by feeding the coefficients of all elements in each reaction into a system of equations and solving for values that would satisfy the system. given that no elements can be added or lost. This results in: $(4/9)TiNb + O_2 \rightarrow (4/9) TiO_2 + (2/9) Nb_2O_5$ and $(4/9)Ti + (4/9) Nb + O_2 \rightarrow (4/9) TiO_2 + (2/9) Nb_2O_5$. The heat of formation for each reaction (without the balancing coefficients) was retrieved and multiplied by the relevant coefficients. Then the heat of formation of the alloy reaction was divided by the naive reaction, which yielded Equation (6); then, the calculation of Equation (7) became trivial. The same process was repeated for the ternary alloys.

The convolutional neural network (CNN) predictor was developed using Tensorflow [29], and written in Python as a Jupyter Notebook. Python version 3.10.5 and Tensorflow version 2.9.1 were used. The dataset of atomic properties for all elements was taken from ref. [30], with nine atomic properties used as predictors: atomic mass, boiling point, density, melting point, electron affinity, Pauling electronegativity, first ionization energy, atomic symbol, and atomic number. For each alloy, a list of the properties of its constituent elements was generated and weighted according to the stoichiometry of the elements in the alloy. At the end of the list, the calculated metric was added.

Once these lists had been generated for all alloys the dataset was ready for the CNN process. The data was randomly split into training, validation, and testing sets (with the same sets being used for all final reported trainings). All data was normalized and then split into categorical and numerical sections. All data was numerical except for the atomic symbols. For each cycle or "epoch" of the model's predictions an "average loss function" was referenced. This function simply took the average of the absolute difference between the metric value predicted by the model and the real calculated "target" value provided. The number of epochs passed was then plotted against the values of the average loss function for each epoch. Based on this plot the ideal number of epochs, 800, was selected for the learning process, which minimized loss but avoided overfitting. The values predicted by the model with different combinations of hidden, densely connected neural network layers (tf.keras.Layers.Dense) with non-linear 'relu' activation were compared with the true values for accuracy (as defined by the average loss function) for different layering schemes. All schemes ended in a Dense (1) output layer with linear activation. The layering schemes tried were: (32,32,32), (64), (64,64), (64,64,64), (128), and (128,128). The layering scheme (64,64), with two hidden layers of 64 neurons each, was determined to yield the lowest average loss for the binary alloys. Repeating this for the ternary alloys revealed an ideal layering scheme of (128,128).

Using the trained CNNs, the values for a list of superconductors were computed. The list of superconductors was taken from ref. [31] and digitized by hand.

Samples presented in Figure 1 were prepared by arc-melting and subjected to several furnace heating cycles in air to obtain an oxide coating. Stoichiometric amounts of elemental aluminum (Kurt J Lesker, Jefferson Hills, PA, USA, 99.99% shots), manganese (Kurt J.

Lesker, 99.95% pieces), nickel (Alfa Aesar, Aesar, UK, 99+% foil), copper (Strem Chemicals, Newburyport, MA, USA, 99.9% foil), yttrium (Strem Chemicals, 99.9% powder), palladium (J&J Materials, Neptune, NJ, USA, 99.95% ingot), tin (Beantown Chemical, Hudson, NH, USA, 99.5% shots), platinum (J&J Materials, Neptune, NJ, USA, 99.95% powder), and gold (APMEX, Oklahoma City, OK, USA, 99.99% bullion) were weighed and combined in 750 mg samples. Sample mixtures containing yttrium were weighed and prepared in an argon filled glovebox to protect against oxidation. These mixtures were transferred to the arc-melting chamber in a closed argon filled vial.

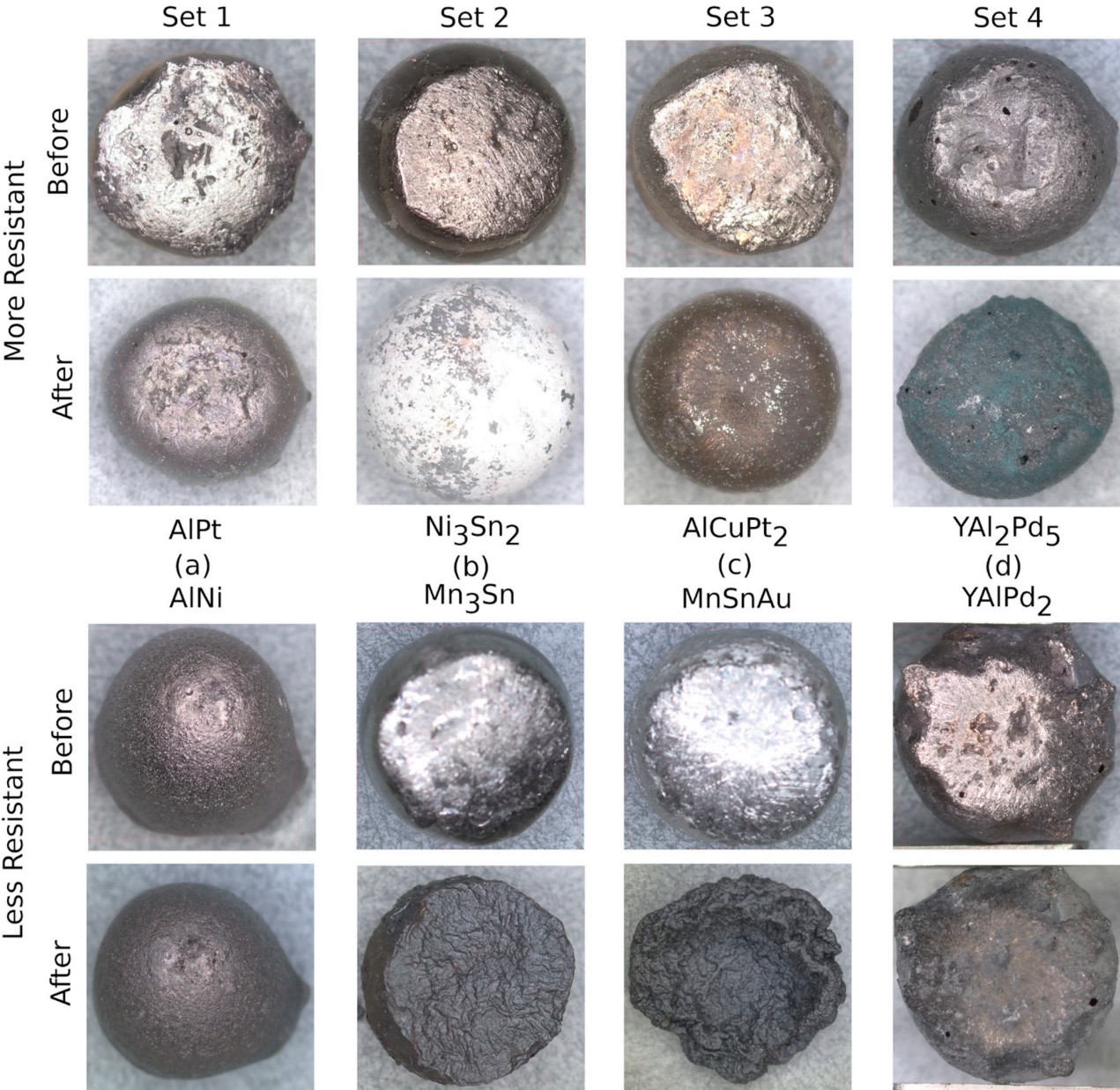

**Figure 1.** Optical images of surfaces of paired binary and ternary compounds, comparing predicted oxidation resistance to that found experimentally for (**a**) AlPt-AlNi, (**b**) $Ni_3Sn_2$-$Mn_3Sn$, (**c**) $AlCuPt_2$-MnSnAu, and (**d**) $YAl_2Pd_5$-$YAlPd_2$. "Before" and "After" refer to before and after controlled oxidation, respectively (see Section 2). In general, trends within each matched set are in good agreement, with the "less resistant" alloy showing more oxidation, with the exception of the Pd-containing ternaries (set 4).

Before melting, the arc-melter was purged and pumped with argon 5 times, and the samples were melted 3 times with flipping between each melt to ensure homogeneity. The samples were placed into a preheated furnace at 300 °C for 5 min each, then were removed from the furnace and allowed to cool in air on the benchtop. Ingot masses were then measured after cooling. This process was repeated for temperatures 500, 700, 900, and 1100 °C, ensuring the furnace was preheated to each desired temperature before each heating cycle. Optical images and measurements were taken on a Zeiss Stemi 508 optical microscope before and after the furnace heating process. Surface areas were estimated through optical measurements of the ingot dimensions, assuming an ellipsoidal shape using the Zeiss Zen software.

AuIn$_2$ and Mo$_3$Re were prepared similarly by arc-melting, from stoichiometric quantities of gold (APMEX, Oklahoma City, OK, USA, 99.99% bullion), indium (Alfa Aesar, 99.999% shot), molybdenum (Alfa Aesar, 99.99% foil), and rhenium (Strem, 99.99% shot). A portion of each as-made button was placed inside a pre-heated furnace at 500 °C (chosen as a reasonable example of an upper limit of temperature in fabrication processes) in air for 5 h, and then removed and imaged.

## 3. Results and Discussion

### 3.1. Defining and Testing an Oxidation Metric

The oxidation of a metal surface comes from a plethora of processes involving adsorption and dissociation of oxygen at the interface, metal-oxygen bond formation, bond rearrangements, and diffusion of oxygen through the oxide scale and at grain boundaries, to result in further oxidation. This complexity means that a single number or unit is incapable of capturing all of the detail regarding metal oxidation. However, the energy scales of these processes, and hence the propensity for formation of various surface oxides, ultimately depend on the differences between metal–metal, metal–oxygen, and oxygen–oxygen bond energies. Since we are looking for multicomponent metal compounds and alloys with superior oxidation resistance compared to the individual elements, intuition suggests that a useful metric is how much metal–metal bonding in the alloy stabilizes those interactions relative to the metal oxides. In particular, the ratio of the heat of formation of the oxide of an alloy to the heat of formation of the oxides formed from a "naive mixture" of the alloy's elements with oxygen should quantify the degree of stabilization.

Consider the trifecta of reactions:

$$A + B \rightarrow AB \;\; \rightarrow \;\; \Delta H_1 \tag{1}$$

$$A + O_2 \rightarrow AO_2 \;\; \rightarrow \;\; \Delta H_2 \tag{2}$$

$$\frac{4}{5}B + O_2 \rightarrow \frac{2}{5}B_2O_5 \;\; \rightarrow \;\; \Delta H_3 \tag{3}$$

These correspond to the formation reactions for the binary alloy *AB*, and the simple oxides of the elements. Intuitively, if reaction (1) becomes infinitely exothermic (i.e., $\Delta H_1 \rightarrow -\infty$), then no oxide should be expected to form on the alloy *AB*. Conversely, if the formation of the alloy *AB* is endothermic, then the bonding in the alloy is destabilized relative to the elements, and oxide formation is expected to be more favorable.

This can be quantified by comparing the heats of formation of the following pair of reactions:

$$\frac{4}{9}A + \frac{4}{9}B + O_2 \rightarrow \frac{4}{9}AO_2 + \frac{2}{9}B_2O_5 \;\; \rightarrow \;\; \Delta H_4 = \frac{4}{9}\Delta H_2 + \frac{5}{9}\Delta H_3 \tag{4}$$

$$\frac{4}{9}AB + O_2 \rightarrow \frac{4}{9}AO_2 + \frac{2}{9}B_2O_5 \;\; \rightarrow \;\; \Delta H_5 = \Delta H_4 - \frac{4}{9}\Delta H_1 \tag{5}$$

In particular, the degree to which bonding in the alloy stabilizes it relative to the parent oxides is given by the ratio:

$$\frac{\Delta H_5}{\Delta H_4} = \frac{\Delta H_4 - \frac{4}{9}\Delta H_1}{\Delta H_4} = 1 - \frac{4}{9}\frac{\Delta H_1}{\Delta H_4} \tag{6}$$

This number takes on a value of one when alloy formation does not change the energetics relative to elements, a value greater than one if the alloy is destabilized relative to oxidation, and a value of less than one if the alloy is stabilized relative to elemental oxidation. Note that the definition of Equations (2)–(5) must include appropriate coefficients to be normalized to one mole of oxygen gas, and that the coefficient on the $\Delta H_1$ term depends on the stoichiometries of the oxides and of the metal alloy. This leads to the following oxidation resistance metric:

$$metric = 1 - \frac{\Delta H_5}{\Delta H_4} = \frac{4}{9}\frac{\Delta H_1}{\Delta H_4} \tag{7}$$

where, again, the coefficient depends on the stoichiometries of the oxides and of the metal alloy. This takes on a number greater than zero when oxidation resistance is increased, and a number less than zero when oxidation resistance is decreased, relative to the elements.

In order to compute this metric on a wide range of materials, we took heats of formation from the Materials Project [28]. All training and computations were done with the value defined in Equation (6), as described in the Section 2, with final plots and interpretation done by post-conversion of those values to the metric, as defined in Equation (7).

To experimentally assess the validity of this metric as a measure of oxidation resistance for the purposes of this study, we synthesized intermetallic compounds spanning the range of predicted parameters (Table 1). We specifically chose pairs of compounds with similar chemical makeup but distinct metric predictions in order to judge the efficacy of the metric. The results are summarized in Table 1 and Figure 1.

**Table 1.** Compositions, oxidation metric, and experimentally measured oxidation quantity (see Section 2), for different matched sets.

|  | Compound | Metric | Oxide (mg/mm$^2$) |
|---|---|---|---|
| **Set 1** | AlPt | 0.2017 | $7.4 \times 10^{-5}$ |
|  | AlNi | 0.1298 | $6.0 \times 10^{-4}$ |
| **Set 2** | Ni$_3$Sn$_2$ | 0.0769 | $1.5 \times 10^{-4}$ |
|  | Mn$_3$Sn | $-0.1057$ | $1.3 \times 10^{-2}$ |
| **Set 3** | AlCuPt$_2$ | 0.2372 | $8.8 \times 10^{-4}$ |
|  | MnSnAu | $-0.1872$ | $8.5 \times 10^{-2}$ |
| **Set 4** | YAlPd$_2$ | $-0.2570$ | $3.2 \times 10^{-3}$ |
|  | YAl$_2$Pd$_5$ | 0.2235 | $6.2 \times 10^{-2}$ |

Considering first the pair of alloys AlNi and AlPt, both were predicted to be stabilized against oxidation, with AlNi the less resistant of the two. After controlled oxidation, the quantity of surface oxide formed was found to be $6.0 \times 10^{-4}$ mg/mm$^2$ and $7.4 \times 10^{-5}$ mg/mm$^2$, respectively. These values were consistent with visual inspection of the surfaces before and after oxidation (Figure 1a) and showed that both were reasonably resistant to oxidation, with AlPt being the higher performer.

The second paired set was Ni$_3$Sn$_2$ and Mn$_3$Sn, with the former predicted to be stabilized against oxidation, and the latter destabilized towards oxidation. These were consistent with visual observations (Figure 1b) and with the measured quantity of oxide formed, which was nearly 100× larger for Mn$_3$Sn than Ni$_3$Sn$_2$.

The third paired set was $AlCuPt_2$ and $MnSnAu$. Both contain elements thought to be resistant to oxidation (Pt and Au). The metric of Equation (7) predicts the former should be much more oxidation-resistant than the latter. This is again what was found in experiment, Figure 1c and Table 1, with again a nearly 100x difference in oxide formation.

The limits of the metric of Equation (7) were demonstrated by the fourth paired set, $YAlPd_2$ and $YAl_2Pd_5$. Here, the metric predicted the former should be less oxidation-resistant than the latter, yet the converse was found experimentally (Figure 1d). One possible reason for this discrepancy is that, compared to all of the other compounds tested here, these are the only ones to contain yttrium, an element whose oxide is known to rapidly hydrolyze in the presence of moisture. We speculate that this additional effect prevents the formation of a protective surface oxide and dominates over the oxide formation in determining ultimate oxide stability.

### 3.2. Predicting the Oxidation Metric from Elemental Compositions

Computing the metric of oxidation resistance described by Equation (7) requires knowledge of the heats of formation of metal oxides and the metal compounds/alloys. While this is possible for materials whose structures are known and whose heats of formations have been computed at a uniform level of theory (such as the values from materials project), there are many superconductors for which such thermodynamic data is not available, either experimentally or computationally. Further, especially when new compounds or alloys are proposed, the structural details are unknown. It is thus desirable to be able to compute this metric without knowledge of all the heats of formation.

To accomplish this, we trained a convolutional neural network (CNN) to predict the value of Equation (6), from which it is simple to extract the desired metric. The CNN is provided with the alloy chemical makeup and stoichiometry, with associated atomic data for each element in the makeup (see Section 2). For initial demonstration that predicting these values is feasible with a CNN, we utilized nine atomic descriptors including atomic number, density, and melting and boiling points (see Section 2). We used the outputs of the Materials Project data-mining as true values for the training process.

The results for binaries and ternaries are shown in Figure 2a,b, respectively. In both cases, the CNN model was able to reproduce the expected values with high fidelity, with an average loss of 0.29% and 0.32%, respectively. This agrees with chemical intuition that the propensity of elements to form bonds with each other versus oxygen should be driven primarily by atomic properties.

To explore in more detail which atomic properties are most important for yielding effective predictions, we trained separate CNNs to predict the oxidation metric using only subsets of the atomic properties as inputs, including density (Figure 3a), atomic mass (Figure 3b), density and atomic mass (Figure 3c), and density and boiling point (Figure 3d). All of these CNN models do a reasonable job of predicting the metric values near zero, but there are differences. The models using just a single predictor show non-linear, non-random deviations at the extrema. These deviations are much reduced when two predictors are used. We thus conclude, in agreement with chemical intuition, that it is primarily the identity of the element that drives bonding behavior. It would be interesting for future work to utilize the emerging methods of explainable machine learning [32,33] to verify this assertion through direct analysis of the machine learning internals.

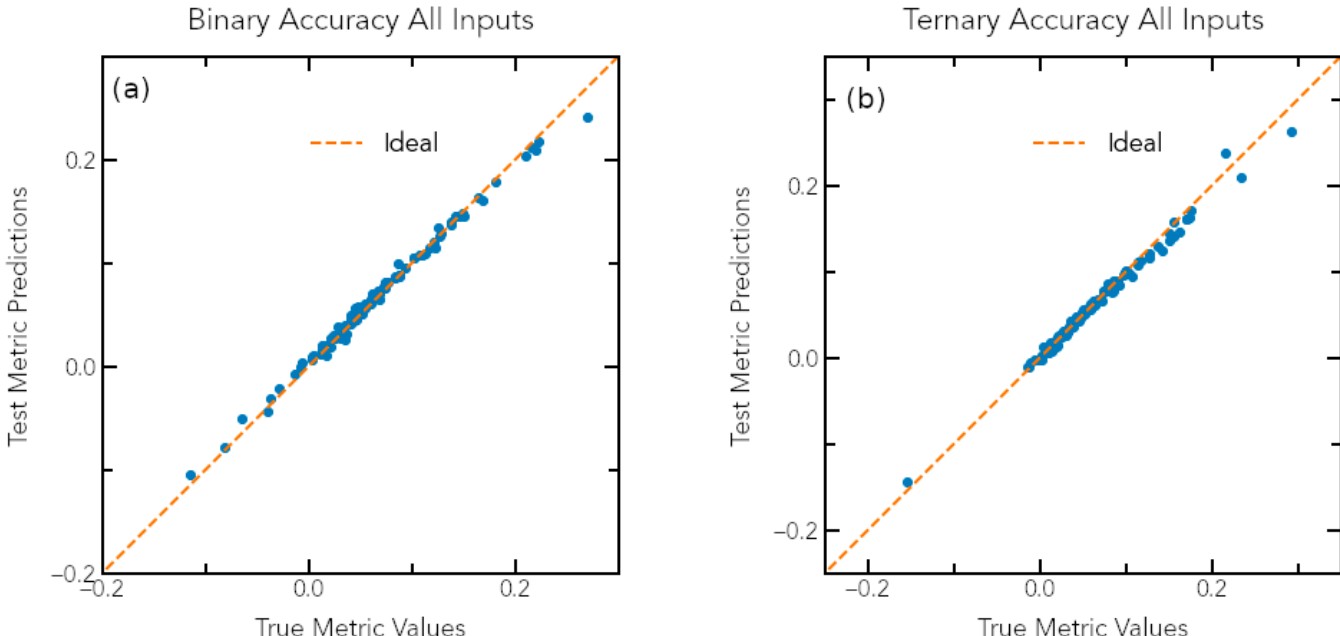

**Figure 2.** Predicted and true values of the metric defined in Equation (7) for (**a**) binary and (**b**) ternary alloys, demonstrating that the trained CNN was capable of predicting values based solely on compositions and atomic input parameters. The ideal line is a line of slope one and intercept zero—the closer to this line, the closer the CNN is properly predicting metric values.

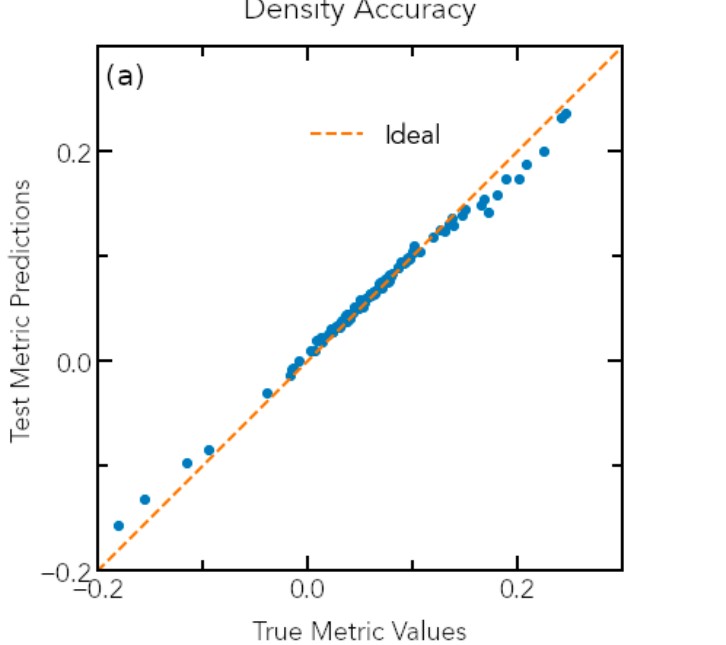
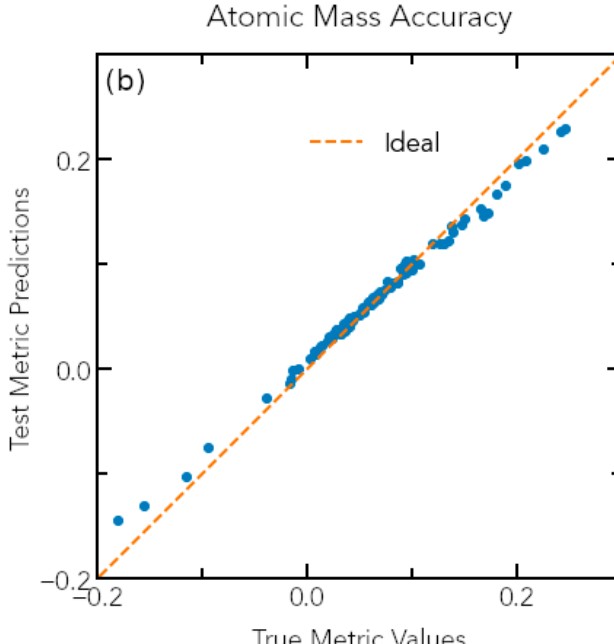

**Figure 3.** *Cont.*

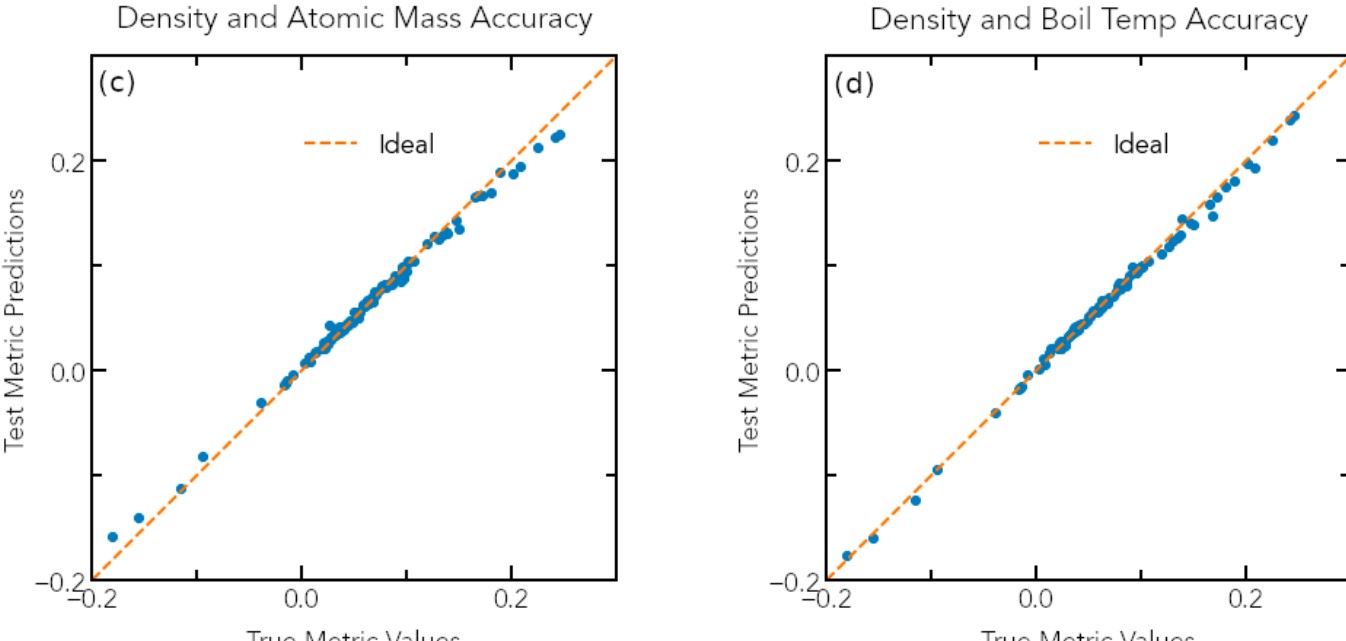

**Figure 3.** By restricting the input predictors, we were able to determine the atomic parameters that, combined with composition, are necessary for the CNN to effectively predict the metric defined in Equation (7). (**a**) Density only; (**b**) Atomic mass only; (**c**) Density and Atomic Mass; (**d**) Density and Boiling Point. The minimal set to produce predictions as good as the full model requires two parameters, e.g., density and boiling point. The ideal line is a line of slope one and intercept zero—the closer to this line, the closer the CNN is properly predicting metric values.

### 3.3. Candidate Superconductors to Enhance QISE

The salient metric for QISE applications is the achievable coherence time ($T_2$) of a full device. This depends not just on the interfaces and oxides, but also on other parameters of the superconducting state. The critical temperature, $T_c$, is particularly important, as devices should be operated at $T \ll T_c$ in order to suppress fluctuations that are detrimental to $T_2$. We thus constructed the overall plot shown in Figure 4, which plots oxidation metric versus $T_c$. Ideal materials are located in the upper right region of this diagram, possessing both a high $T_c$ and a high resistance to oxidation. Many compounds are predicted to exist at the frontier, i.e., most oxidation-resistant, with one of the most promising being $PbTa_3$ with a $T_c = 17$ K and an oxidation metric of +0.0027.

The oxidation metric was predicted using the CNN, with the compounds themselves and $T_c$'s taken from an experimental list of known intermetallic superconductors. Compared to all known binary and ternary alloys in the Materials Project, the range of accessible oxidation metrics here is much smaller: $-0.0113$ to $+0.0027$. Nonetheless, we chose a pair of predicted candidates, $Mo_3Re$ ($-0.006$) and $AuIn_2$ ($+0.001$), to evaluate for oxidation resistance. Both showed no detectable change in mass after controlled oxidation at 500 °C, indicating general robustness against oxidation. Visual inspection (Figure 2 insets) showed that the former appeared to have a uniform thin film of an oxide, indicated by a light yellowing of the surfaces, while the latter showed regions of no apparent oxide, and regions with a blue coloring. These are qualitatively consistent, with the latter being slightly more oxidation-resistant than the former. It is worth noting that though the critical temperature of $AuIn_2$ is low (0.20 K), its oxidation resistance allows for the possibility of other useful applications. For instance, microwave single photon detectors, which can be implemented both on the basis of SIS junctions, where a thin and uniform oxide barrier is important [34]. It could also be implemented on the basis of calorimeters with extremely low critical temperature such as Hf and Ir [35].

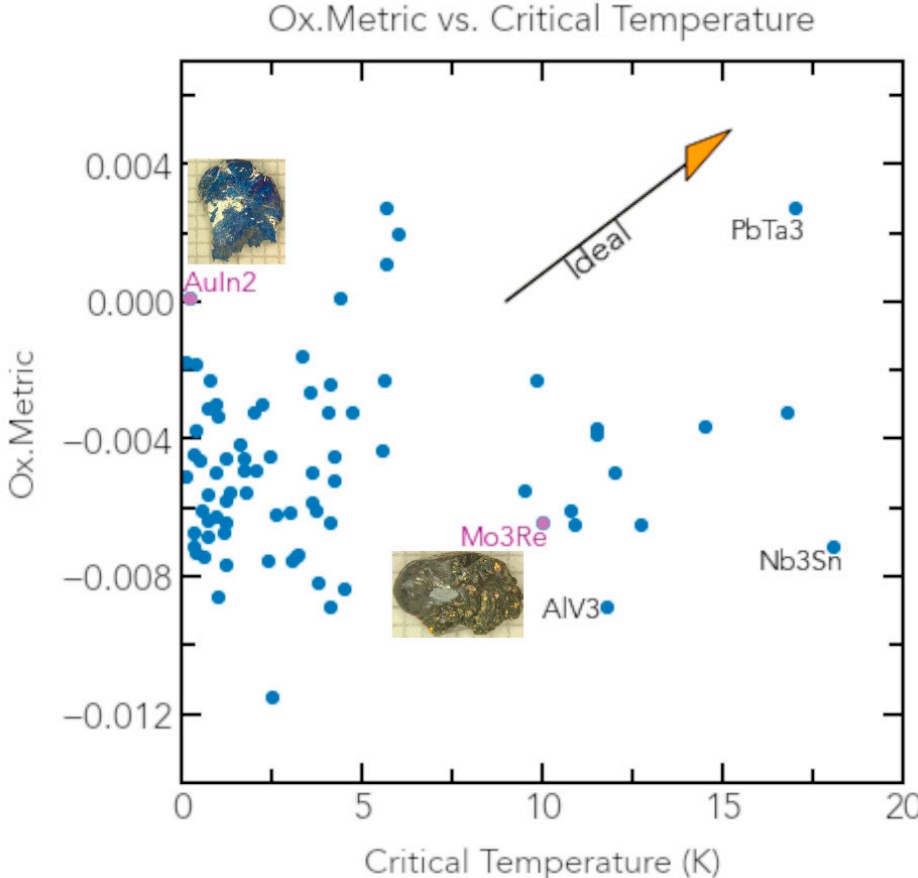

**Figure 4.** Combining the oxidation metric with $T_c$ allows identification of promising superconductors for QISE applications. The insets show the results of trial oxidation experiments on two predictions, $AuIn_2$ and $Mo_3Re$.

## 4. Conclusions

Here we reported the development of a thermodynamic metric to rank a material's quality with respect to the formation of surface oxides, and assessed the quality of this metric by experimentally preparing a set of metal alloys with subsequent controlled oxidation. We then trained a convolutional neural network (CNN) to predict the value of this metric solely from atomic properties and stoichiometries and applied it to compute the metric for known intermetallic superconductors to identify superconductors with a high figure of merit for QISE applications and checked some of these predictions with initial experimental synthesis and oxidation screening.

Our results demonstrate a need to discover superconductors that are more resistant to oxidation than known families, a task that should be possible given the much wider range of oxidation resistance exhibited by known binary and ternary compounds compared to known superconductors. Further, our approach lays the foundation for a broader materials discovery pipeline to improve superconductors for QISE applications, perhaps in combination with recent advances in multiproperty predictions via "AI/ML" approaches [36–38]. We anticipate that a combination of these approaches with incorporation of predicted materials into full QISE devices will lead to the next revolution in superconducting qubit performance, as well as enabling improved single-photon detectors [34,35] and related technologies.

**Author Contributions:** Conceptualization, T.M.M.; methodology, C.K., B.W., A.I. and T.B.; software, C.K.; validation, T.M.M.; investigation, C.K., B.W., A.I., E.H. and T.B.; writing—original draft preparation, C.K., B.W., A.I., E.H., T.B. and T.M.M.; writing—review and editing, C.K., B.W., A.I., E.H., T.B. and T.M.M.; visualization, C.K.; supervision, T.M.M.; project administration, T.M.M.; funding acquisition, T.M.M. All authors have read and agreed to the published version of the manuscript.

**Funding:** This work was funded by the U.S. Department of Energy, Office of Science, National Quantum Information Science Research Centers, Co-Design Center for Quantum Advantage (C2QA) under contract number DE-SC0012704. TBe was supported by the NSF-MRSEC through the Princeton Center for Complex Materials, Award #DMR-2011750.

**Data Availability Statement:** The data presented in this study are available within the article. We are working to make the codes underlying this study available on Github.

**Conflicts of Interest:** The authors declare no conflict of interest.

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
