# Peer review of "Machine-Guided Design of Oxidation-Resistant Superconductors for Quantum Information Applications"

_inorganics, doi:10.3390/inorganics11030117_

Round 1

Reviewer 1 Report

The work done by Koppel et. al. reported using machine learning to find new superconductors. The cnn method was used. The work is well presented and organized logically well. The work should be published.

Author Response

We thank the referee for the positive view of our work.

Reviewer 2 Report

This manuscript presents investigations of oxidation properties of various superconducting materials for qubit applications. The paper is interesting and sound and can be accepted for publication after the authors will correct the manuscript following the comments below.
1. Usually, the main material for superconducting qubits is Al. Since in the introduction the authors write about niobium and compare it with tantalum, they should also compare with aluminum and discuss its advantages and disadvantages for qubit applications.
2. The authors consider AuIn2, which has very low critical temperature. First, the concrete value of Tc is not clearly visible at the plot in Fig. 4, please, give this value explicitly in the text. Second, it is not clear from the text whether this value measured by the authors, or taken from the literature, or just calculated. On the other hand, this low critical temperature makes this material unsuitable for qubit applications. But, it is quite suitable for microwave single photon detectors, which can be implemented both on the basis of SIS junctions, where thin and uniform oxide barrier is important, see [npj Quantum Information 8, 61 (2022) https://doi.org/10.1038/s41534-022-00569-5], and also on the basis of calorimeters with extremely low critical temperature such as Hf and Ir, see [Supercond. Sci. Technol. 35 105013 (2022) https://doi.org/10.1088/1361-6668/ac8a24]. I think that adding these citations in the manuscript will enhance its motivation.
3. "a list of the lists" and "the list of lists" in page 3 looks strange. I would expect just: "For each alloy a list of the properties"...

Author Response

We thank the referee for the positive view of our work, and the comments to help improve the manuscript.

We agree with the referee that Al predates the use of Nb or Ta in superconducting qubits, and is still used even in Nb/Ta based qubits for the josephson-junction element. We have added discussion to the introduction and motivation to include information on Al-based qubits.

We thank the referee for this excellent suggestion. We have included additional language and references to describe these possibilities.

Our terminology here was shaped by the details of the implementation within python. As the code will be released as part of publication, we have simplified the language within the manuscript to avoid confusion.

Reviewer 3 Report

The authors present a machine-guided design of oxidation resistant superconductors for quantum information applications. While the part related to the materials design is described in details, the link with the quantum information applications is not clearly motivated and related to the materials design. I suggest the authors to explain why the analysis about the design of the superconductors is relevant for the quantum technologies problems. Such points should be clarified in the introduction, discussion and conclusions. This aspect is important and has to be addressed before the manuscript can be published.

Author Response

Our additions to address the comments by referee #2 partially address this. We have also more explicitly stated how materials design is necessary to make step change improvements in qubit coherence time, which has an immediate benefit for all QIS applications.